# Adipokines as Diagnostic and Prognostic Markers for the Severity of COVID-19

**DOI:** 10.3390/biomedicines11051302

**Published:** 2023-04-27

**Authors:** Thomas Grewal, Christa Buechler

**Affiliations:** 1School of Pharmacy, Faculty of Medicine and Health, University of Sydney, Sydney, NSW 2006, Australia; thomas.grewal@sydney.edu.au; 2Department of Internal Medicine I, Regensburg University Hospital, 93053 Regensburg, Germany

**Keywords:** adiponectin, chemerin, leptin, SARS-CoV-2, COVID-19, pneumonia, intensive care

## Abstract

Accumulating evidence implicates obesity as a risk factor for increased severity of disease outcomes in patients infected with severe acute respiratory syndrome coronavirus type 2 (SARS-CoV-2). Obesity is associated with adipose tissue dysfunction, which not only predisposes individuals to metabolic complications, but also substantially contributes to low-grade systemic inflammation, altered immune cell composition, and compromised immune function. This seems to impact the susceptibility and outcome of diseases caused by viruses, as obese people appear more vulnerable to developing infections and they recover later from infectious diseases than normal-weight individuals. Based on these findings, increased efforts to identify suitable diagnostic and prognostic markers in obese Coronavirus disease 2019 (COVID-19) patients to predict disease outcomes have been made. This includes the analysis of cytokines secreted from adipose tissues (adipokines), which have multiple regulatory functions in the body; for instance, modulating insulin sensitivity, blood pressure, lipid metabolism, appetite, and fertility. Most relevant in the context of viral infections, adipokines also influence the immune cell number, with consequences for overall immune cell activity and function. Hence, the analysis of the circulating levels of diverse adipokines in patients infected with SARS-CoV-2 have been considered to reveal diagnostic and prognostic COVID-19 markers. This review article summarizes the findings aimed to correlate the circulating levels of adipokines with progression and disease outcomes of COVID-19. Several studies provided insights on chemerin, adiponectin, leptin, resistin, and galectin-3 levels in SARS-CoV-2-infected patients, while limited information is yet available on the adipokines apelin and visfatin in COVID-19. Altogether, current evidence points at circulating galectin-3 and resistin levels being of diagnostic and prognostic value in COVID-19 disease.

## 1. Introduction

Severe acute respiratory syndrome coronavirus type 2 (SARS-CoV-2) infection causing Coronavirus disease 2019 (COVID-19) is characterized by a plethora of very heterogeneous symptoms and unpredictable disease trajectories. While there are cases with asymptomatic infections and patients with mild symptoms, such as dry cough, fever, and dyspnoea, a substantial number of SARS-CoV-2 infections lead to serious and life-threatening outcomes with lobar pneumonia affecting multiple lopes and acute respiratory distress syndrome (ARDS) [1]. The underlying causes responsible for the differential outcome of SARS-CoV-2 infections remain to be fully elucidated, but common factors that contribute to disease severity include vaccination state, age, intake of immunosuppressive drugs, and also metabolic disorders [2].

It is now well documented that comorbidities, particularly cardiovascular disease, metabolic-associated fatty liver disease, diabetes, and hypertension, increase the risk of poor outcomes in COVID-19 patients. Consistently associated with these metabolic diseases are chronic inflammation, impaired immunity, coagulopathy, and disturbed glucose homeostasis, all of which contribute to COVID-19 severity [2,3,4].

Respiratory infections are mostly caused by viruses [5], and many reports recognized that obesity contributes to a more severe outcome in respiratory diseases triggered by viral infections. A meta-analysis uncovered that in patients infected with H1N1 influenza virus, extreme obesity (body mass index (BMI) ≥ 40 kg/m^2^) was associated with a higher risk of intensive care unit admission or death [6]. Along these lines, another study found that both underweight as well as morbidly obese patients infected with influenza virus were hospitalized more often [7]. Similarly, obesity-associated metabolic diseases, such as type 2 diabetes or hypertension, were related to a worse outcome in patients infected with Middle East respiratory syndrome coronavirus (MERS-CoV) [8].

The causal relationship between obesity and poor disease outcome upon viral infection are not fully understood, but in recent years, there have been some critical insights. In individuals with adiposity, many observations have linked the higher risk for metabolic and cardiovascular diseases with increased inflammation and an impaired immune response [9]. Moreover, as outlined in more detail below, this dysregulation of the immune response in obese patients is now recognized to substantially contribute to an increased severity of pulmonary infections. Obesity is associated with an increased number and inflammatory capacity of monocytes, macrophages, neutrophils, and CD4 T cells. In particular, the pro-inflammatory state of neutrophils and macrophages in obese patients greatly contributes to an increased expression of cytokines and elevated levels of reactive oxygen species [10]. On the other hand, decreased effector functions of eosinophils, natural killer cells, dendritic cells, CD8 T cells, and B cells in obese patients have also been observed [10]. De-regulated T cell function contributing to disease severity is exemplified by CD4 and CD8 T cells from overweight and obese subjects exerting an impaired response to H1N1 influenza virus infection compared to CD4 and CD8 T cells of normal-weight patients [11].

In addition to influenza, MERS-CoV and various other infectious diseases, obesity also negatively impacts the outcome of COVID-19 [2,5]. This observation was recently confirmed in a meta-analysis of 34,390 patients from 12 different studies identifying the association of obesity with poor outcome, mortality, and severity in COVID-19 patients [12]. Likewise, several other studies identified obesity as a risk factor for worse disease outcomes of COVID-19, suggesting that higher BMI may be useful for risk stratification [13,14,15].

Several features accompanying obesity seem to provide a microenvironment that reflects clinical attributes observed in SARS-CoV-2-infected patients. Obesity is associated with higher plasma levels of interleukin-6 (IL-6), C-reactive protein (CRP), and lipopolysaccharide (LPS), and accordingly, driving mononuclear cells into a pro-inflammatory (M1) state [16]. In addition, obese individuals are commonly characterized by an increased number of circulating neutrophils, which are further elevated in patients with comorbidities (Figure 1). These neutrophils are activated, favouring the production of reactive oxygen species and pro-inflammatory cytokines [17]. Weight loss is associated with a decline in the neutrophil-to-lymphocyte ratio [18]. Striking similarities to these obesity-related inflammatory traits can also be observed in SARS-CoV-2-infected patients. In COVID-19, IL-6, CRP, and LPS levels are elevated and associated with adverse clinical outcomes [19,20,21]. Severe COVID-19 is characterised by an increased neutrophil count and lymphocyte depletion, and the neutrophil-to-lymphocyte ratio positively correlates with disease severity [22]. Pro-inflammatory M1 macrophages prevail in COVID-19 patients, and their polarization into M2 cells is disturbed [23]. Finally, patients have a higher risk for severe COVID-19 disease outcome if comorbidities such as diabetes, hypertension, and cardiovascular diseases, which are all more prevalent in the obese, exist [24,25] (Figure 1).

Further insights on the association of obesity and COVID-19 come from epidemiologic studies, which strongly suggest that the distribution of body fat is decisive for disease outcome of SARS-CoV-2-infected patients [9]. Visceral adipose tissue surrounding inner organs not only differs from subcutaneous fat depots with respect to its localization in the body, but these two fat depots are also greatly dissimilar in regard to their cellular composition, fat storage, and capacity to secrete adipokines [26,27]. Visceral adipose tissues produce about two- to three-fold more IL-6 in comparison to subcutaneous fat, and portal vein IL-6 levels correlate with the amounts of CRP, an established clinical marker of inflammation [28]. This suggests that IL-6 released from visceral fat directly impacts the hepatic production of acute phase proteins such as CRP. IL-6 levels in the portal vein being higher compared to those in the hepatic vein indicates that IL-6 derived from visceral fat is eliminated by the liver [29]. In line with this, obesity is often characterized by low-grade systemic inflammation, and the visceral fat area is independently associated with white blood cell count and high-sensitivity CRP (hs-CRP) levels [30].

These observations extend to the association of specific adipose tissue allocations with COVID-19 severity reported in several studies. Abdominal fat distribution defined by higher visceral fat mass and comparably lower subcutaneous adipose tissue mass had an increased risk of intensive care unit admission for COVID-19, and this was independent of BMI [31]. Furthermore, patients with severe COVID-19 had a higher visceral fat area, while the area of subcutaneous adipose tissue mass remained unaffected [32]. Likewise, the adjusted hazard ratio for severe COVID-19 outcome of hospitalized patients for high versus low visceral adipose tissue was 1.97, pointing at high visceral adipose tissue being associated with more severe disease [33]. Additionally, genome-wide association studies revealed that patients with genetically predicted visceral adiposity had a higher risk of infection with SARS-CoV-2, hospitalization, and severe outcome. These latter associations were significant also after adjusting for BMI [34]. Yet, the use of BMI for risk assessment of disease outcome after SARS-CoV-2 infection may be limited, as BMI correlated with total fat mass and subcutaneous and visceral fat mass, though the association of BMI with visceral fat was the lowest [35].

Excessive accumulation of fat disturbs adipose tissue function. In the obese, consequences include low-grade inflammation, adipose tissue fibrosis, as well as elevated release of proteins from adipose tissue (adipokines) into the circulation [36]. Adipokines exert diverse functions and regulate appetite, fertility, glucose, lipid homeostasis, and, most relevant for the COVID-19 disease trajectory, immunity [37,38,39]. Hence, the latter observation led to studies determining systemic levels of different adipokines in COVID-19 patients to find possible associations with disease severity and outcome. Furthermore, directly linking SARS-CoV-2 with fat tissue dysfunction, SARS-CoV-2 can infect adipocytes and adipose tissue macrophages. This not only initiates inflammatory pathways in both cell types, but also decreases adiponectin expression [40,41]. In addition, inflammatory cytokines seem to contribute to adiponectin downregulation [42]. As adiponectin behaves as an anti-inflammatory factor, these multiple pathways downregulating adiponectin levels further drive inflammation. These findings point to a potential relationship between COVID-19 progression and outcome with adipokine production from adipose tissues in obese patients.

Thus, in recent years, researchers have determined chemerin, adiponectin, leptin, resistin, galectin-3, apelin, and visfatin levels in SARS-CoV-2-infected patients. Most interestingly, circulating galectin-3 and resistin levels may have diagnostic and prognostic value in the COVID-19 disease. This review article summarizes and discusses the studies that have measured circulating adipokine levels in COVID-19 patients.

## 2. Adipokines

### 2.1. Chemerin

The chemoattractant adipokine chemerin is highly expressed and secreted from adipocytes and hepatocytes [43]. Once released into the extracellular milieu, the binding of chemerin to the chemokine-like receptor 1 (CMKLR1) can induce the migration of leukocytes. Chemerin is secreted as an inactive protein and to function as a chemoattractant for monocytes, dendritic cells, T cells or natural killer cells, chemerin has to undergoe proteolysis at the C-terminus. The proteases that cleave chemerin are mostly serine proteases, and chemerin processing is increased in inflammation [44]. Thus, in inflammatory diseases and infection, chemerin attracts immune cells and contributes to their enrichment at the sites of infection. This feature allows chemerin to participate in the resolution of inflammation, with short C-terminal chemerin peptides also exerting anti-inflammatory effects in vitro and in animal models. This effect was abolished in CMKLR1 null mice, favouring chemerin to regulate an anti-inflammatory response through CMKLR1 activation [45].

In addition, and supporting a role for chemerin in viral clearance, CMKLR1-deficient mice infected with pneumonia virus were characterized by a delayed clearance of the virus, increased infiltration of neutrophils and more severe disease [46]. It should be noted that CMKLR1 is also a receptor for resolvin E1 and possibly resolvin E2 [47], allowing for alternative explanations, as resolvins derived from polyunsaturated fatty acids act as pro-resolving lipids to initiate the resolution of inflammation [47]. Indeed, administration of resolvin E1 lowered the ocular inflammatory response upon infection with herpes simplex virus [48], and the levels of E-series resolvins derived from eicosapentaenoic acid were changed in COVID-19 [49].

The regulation of the chemerin-mediated chemotaxis of immune cells is complex and not fully understood, as the multiple proteases facilitating chemerin processing generate different isoforms, several of those with biological activity [38,43]. In addition, besides CMKLR1, chemerin also binds with high affinity to atypical chemokine CC motif receptor-like 2 (CCRL2) and G protein-coupled receptor 1 (GPR1). GPR1 is also believed to facilitate chemerin-inducible chemotactic behaviour, while CCRL2 is a non-signalling receptor and seems to increase local chemerin concentrations [43,50]. Chemerin activity is usually monitored by beta-arrestin 2 Tango and calcium mobilization assays [43,51,52,53]. Hence, yet unidentified receptors or differential activities of chemerin isoforms that are not detected by these assays may exist. Notably, when examining a panel of overlapping synthetic chemerin peptides, chemerin-derived peptides within the chemerin protein mimicked the antimicrobial effects of longer chemerin isoforms [54]. Thus, chemerin isoforms not regarded as active may well exert yet unidentified biologic functions, and this may also apply to relatively short chemerin peptides.

In the serum of healthy humans, C-terminally unprocessed chemerin appeared to be the most prominent isoform, which was inactive when tested using the beta-arrestin 2 Tango assay. Circulating chemerin levels were induced in obesity, with higher levels of C-terminally truncated isoforms. However, these processed chemerin isoforms were also believed to exert no activity [52,55]. Serum chemerin levels correlated with visceral fat area [56] were consistently higher in patients with hypertension and various inflammatory diseases [57], indicating that these previously rated inactive isoforms may have pathophysiological functions.

Despite the association of chemerin upregulation in inflammatory settings, studies monitoring chemerin levels in patients with viral infections are still sparse. For example, chemerin levels were determined in intrauterine infections with cytomegalovirus (CMV), which can cause infant brain damage [58]. Chemerin levels in amniotic fluid were highly predictive of the severity of congenital CMV, and differentiated between foetuses with severe and asymptomatic disease [59]. On the other hand, serum chemerin levels in hepatitis C virus (HCV)-infected patients positively associated with leukocyte count and negatively with markers of hepatic function, yet did not correlate with viral load and remained unaltered even after the efficient elimination of the virus, excluding HCV infection to change the circulating levels of this adipokine [60].

Up to date, three studies determined systemic chemerin levels in the COVID-19 disease. One analysis showed that serum chemerin amounts were lower in 70 patients infected with SARS-CoV-2 at the day of hospital admission compared to 20 healthy controls [61]. While white blood cell count was comparable in the two groups, CRP, ferritin, alanine aminotransferase (ALT), and gamma-glutamyl transferase (GGT) activities were higher and albumin was lower in the infected patients. Total cholesterol, triglyceride, low- and high-density lipoprotein (LDL, HDL) levels were comparable among the two groups [61]. Serum chemerin amounts were not associated with changes in ALT, GGT or ferritin, and did not correlate with COVID-19 severity assessed by the presence of pneumonia, dyspnoea, or intensive care unit admission [61].

In contrast to the findings described above, another study observed that plasma chemerin levels were increased in 88 COVID-19 patients compared to 21 healthy controls. This difference was significant on Day 1, 5, and 14 after hospital admission. Plasma chemerin concentrations correlated with the amount of inflammatory markers CRP and tumour necrosis factor (TNF). Plasma chemerin levels on Day 1 did not differ between patients when classified into non-hospitalized, hospitalized in conventional care unit or hospitalized in intensive care unit [62]. This changed on Day 14 with hospitalized patients displaying higher chemerin levels compared to patients with mild disease. Though serum chemerin amounts on Day 14 did not discriminate between patients admitted or not admitted to the intensive care unit, it was an independent risk factor for death. Plasma chemerin levels were also higher in deceased compared to surviving COVID-19 patients on Day 1, 5, and 14 [62].

Finally, in a larger cohort of 254 SARS-CoV-2-infected patients, serum chemerin levels did not differ between patients with non-severe and severe disease on the day of admission, and when measured 7 and 28 days later. Notably, patients with severe disease had higher BMI and waist circumference, elevated ALT, aspartate aminotransferase (AST) and GGT levels, and an increased neutrophil count, whereas lymphocyte and monocyte number, as well as serum albumin levels were reduced. In patients with moderate or severe acute infection, serum chemerin levels first decreased on Day 7 in comparison to the day of hospital admission and subsequently increased at the 28-day follow-up. As this profile of plasma chemerin levels was not observed in COVID-19 patients with mild disease over this 28-day period, it was suggested that serum chemerin levels are considered as a marker for the resolution of inflammation [63]. However, while randomized clinical trials proved that anti-inflammatory and pro-resolving corticosteroids can improve survival in severe COVID-19 [64], dexamethasone therapy did not affect plasma chemerin levels [62].

In summary, more patient data are still required to clarify if serum chemerin levels differ between healthy controls and SARS-CoV-2-infected patients. Chemerin in the circulation was not related to disease severity and was not prognostic for a worse disease course or survival in two of the three studies published to date (Table 1).

### 2.2. Adiponectin

Adiponectin is a very well-studied adipokine, almost exclusively produced in adipocytes. These cells secrete trimeric, hexameric, and high molecular weight adiponectin complexes. Adiponectin has insulin-sensitizing and cardio- and hepatoprotective activities. Notably, serum adiponectin levels decline in the obese, and low levels of the high molecular adiponectin complexes contribute to metabolic diseases such as insulin resistance and non-alcoholic fatty liver disease [42,91,92].

Adiponectin receptor 1 (AdipoR1), AdipoR2, and T-cadherin are receptors for adiponectin, and ligand binding leads to the activation of a variety of signalling pathways [93]. AdipoR1 and AdipoR2 are mainly responsible for the mediation of the immunoregulatory effects of this adipokine. While AdipoR2 was needed for the effects of adiponectin in M2 macrophage polarization [94] and for the upregulation of IL-1 receptor antagonist in THP-1 monocytes, AdipoR1 was involved in the suppression of TNF and CC-chemokine ligand 2 (CCL2) expression [95]. The anti-inflammatory effects of adiponectin in LPS-stimulated macrophages included the downregulation of IL-6 and induction of IL-10 expression [96]. Along these lines, IL-10 levels were higher in macrophages incubated with recombinant adiponectin [97]. Adiponectin, moreover, inhibited the expression of TNF in macrophages [92]. Peritoneal and adipose tissue-resident macrophages of adiponectin-deficient mice were accordingly polarized to the M1 type, further providing evidence for an anti-inflammatory role of adiponectin [98]. A protective effect of adiponectin was also described in human bronchial epithelium stimulated with TNF or poly(I:C), with physiological concentrations of adiponectin lowering CCL2 and C-X-C motif chemokine ligand 1 (CXCL1) levels. In accordance with the anti-inflammatory role of adiponectin, circulating adiponectin levels were found to negatively correlate with hs-CRP levels [92,99].

However, various studies also observed the pro-inflammatory effects of adiponectin, such as the activation of the transcription factor nuclear-factor kappa B (NF-kappaB) [37]. Here, adiponectin increased NF-kappaB-inducible IL-6, CCL2, CCL20, and CXCL8 expression [100,101]. While it remains challenging to unravel opposite outcomes of adiponectin-mediated receptor signalling, current models indicate that adiponectin-related effects depend on the activation status of the cell and the distinct signalling potential of the different adiponectin isoforms [37,91]. Unexpectedly, high adiponectin plasma levels were detected in several inflammatory diseases such as inflammatory bowel diseases and rheumatoid arthritis [94,102]. Total and high molecular weight adiponectin were reported to be positively related to all-cause and cardiovascular mortality rate across several clinical studies, and only few studies reported an inverse relationship [103]. In patients with acute respiratory failure, higher plasma adiponectin concentrations were associated with mortality [104].

Studies that determined serum adiponectin levels in patients with viral infections are still rare. In mouse models for HCV infection, the efficient elimination of the virus was related to a decline of adiponectin plasma levels [105]. Yet, the anti-viral therapy of HCV had no effect on circulating adiponectin amounts in patients, and this also applied to hepatitis B virus (HBV)-infected patients [106]. An association between low adiponectin levels in the obese and hyper-responsiveness of immune cells to influenza A virus infection has also been suggested [107].

Considering the immune-regulatory effects of adiponectin, the systemic levels of adiponectin were measured in patients with COVID-19. Serum adiponectin levels were lower in 62 hospitalized COVID-19 patients compared to 62 healthy controls. Here, the high molecular weight adiponectin complexes, but not the total serum adiponectin amounts, correlated positively with the level of lung injury, as judged by the lung ultrasound score. A limitation of this study was the higher BMI of COVID-19 patients in this cohort, a feature, whose association with lower circulating adiponectin levels has been well-described [74,91]. Adiponectin was also found to be reduced in the plasma of severely ill COVID-19 patients in comparison to healthy controls and patients with mild symptoms. This study also included severely ill patients not infected with SARS-CoV-2, and plasma adiponectin was also low in this cohort. Hence, critical illness independent of SARS-CoV-2 infection appears linked to reduced systemic adiponectin levels. Notably, the healthy controls were sex, age, and BMI matched to the COVID-19 patients admitted to the intensive care unit. Critically ill COVID-19 patients were generally older, often male, and had a higher BMI compared to patients with mild disease outcome [76].

In a cohort of 254 COVID-19 patients, serum adiponectin amounts were reduced in patients with severe disease compared to those with non-severe disease on the day of admission, and 7 and 28 days later. Alike the drawback of the study described above, the non-severely ill patients had lower BMI and waist circumference than the severely ill patients, and this may contribute to differences in circulating adiponectin levels irrespective of disease severity. Serum adiponectin levels increased in both groups during their 28 days in hospital care [63].

In hospitalized COVID-19 patients, plasma adiponectin amounts were not related to clinical outcomes, and did not change during their stay at hospital [67]. In a group of 145 hospitalized patients, adiponectin levels were not related to intensive care requirement or disease outcome [69]. Systemic adiponectin levels did not differ between patients experiencing mild, moderate or severe COVID-19 disease [66,68]. Moreover, adiponectin in plasma was similar in healthy controls and patients with COVID-19 pneumonia. A significant difference between these two groups was only observed in the subgroup with a BMI of <25 kg/m^2^. Within this subgroup, patients were characterized by lower adiponectin levels [71]. A separate study observed higher serum adiponectin levels in 92 COVID-19 patients compared to healthy controls. Adiponectin levels were not related to ARDS or outcome [108]. A prospective study enrolled hospitalized patients with community-acquired pneumonia (CAP) caused by bacteria or SARS-CoV-2 infection. Body composition and metabolic traits were similar between these two groups. Patients with COVID-19 had increased levels of interferon (IFN)-γ, IL-4, IL-5, and IL-6 compared to patients suffering from bacterial CAP, but CRP and adiponectin levels did not differ between these two groups [70]. Adiponectin was also similar in patients with severe pneumonia caused by COVID-19 or non-COVID-19, and was not related to markers of inflammation or disease outcome [65]. It was observed that the 12 patients with COVID-19 respiratory failure had lower plasma adiponectin amounts as compared to the 17 patients with other causes of respiratory failure [109]. A second prospective study, which included 195 hospitalized COVID-19 patients, could neither observe differences of plasma adiponectin levels between patients not admitted and those admitted to intensive care, nor between survivors and non-survivors [67]. The latter analysis also evaluated associations with gastrointestinal complications and found increased serum adiponectin levels in patients with intestinal lesions, but not in patients with hepatitis [108].

In summary, given the rather inconsistent data, often limited by small cohort numbers or independent factors unrelated to COVID-19, such as BMI, there is currently little evidence for adiponectin as a diagnostic or prognostic marker in COVID-19 (Table 1).

### 2.3. Leptin

Leptin is mostly derived from adipocytes and informs the brain about the fat reserves within the body, and thus, regulates energy expenditure and appetite. In addition, leptin is a pro-inflammatory cytokine and shares structural and functional similarities with IL-6. Both of these cytokines signal through the janus kinase/signal transducer and activator of the transcription (JAK/STAT) pathway [39,110].

Leptin acts by binding to the leptin receptor, which exists as a soluble isoform, isoforms with short cytoplasmic domains, and a long isoform. The latter is expressed almost ubiquitously, including most immune cells, and believed to transduce leptin signals to cells, including the activation of JAK/STAT and also NF-kappaB [39,110,111]. Furthermore, leptin enabled macrophages to undergo M1 polarization, and favoured T cells to differentiate into pro-inflammatory T helper 17 cells characterized by their production of IL-17 at the expense of differentiation to regulatory T cells. Leptin also increased the proliferation of natural killer cells [39,110].

In activated T cells, leptin enhanced the expression of glucose transporter 1 (GLUT1), thereby enabling T cells to cover the increased demand for glucose during the energy-consuming immune response. In humans and rodents, loss or downregulation of leptin impaired T cell proliferation and synthesis of inflammatory cytokines, and increased their susceptibility to infectious diseases. Strikingly, and identifying leptin as a key player controlling the immune system, treatment with recombinant leptin normalized the immune response [39,110,111].

Hyperleptinemia and leptin resistance are characteristics of the obese, and there is strong evidence that inefficient leptin signalling contributes to the higher susceptibility for bacterial and viral infections and the impaired immune response of obese patients [39,110]. For instance, leptin-deficient *ob*/*ob* mice with sepsis had more severe disease and higher mortality compared to the respective controls, and this was improved by leptin replacement. Further studies showed that leptin signalling in the brain improved the systemic immune response and increased survival [112].

Mimicking bacterial infection, animal studies identified that LPS induces plasma leptin levels, as well as the adipocyte leptin mRNA expression [113]. Only few studies in humans have addressed leptin levels in experimental endotoxemia, with intravenous administration of endotoxin not affecting the circulating leptin levels of 12 healthy subjects after 6 and 24 h [114]. In contrast, another study reported a significant rise of leptin 24 h after LPS injection [115]. These opposite findings remain to be clarified, but the latter studies included only females, whereas 10 males and 2 females were enrolled in the study by Bornstein and colleagues [114,115]. Hence, future research will have to evaluate possible sex-specific effects determining the ability of LPS to induce leptin expression.

Indeed, sex-related differences related to leptin and disease outcome triggered by infections may exist, as a prospective study documented high leptin levels at baseline to predispose individuals to sepsis. Increased leptin in the acute phase was related to a more favourable outcome in men, but associated with an elevated risk of death in women [116]. In sepsis patients (56% male), serum leptin levels were higher in survivors [117]. These studies suggested that the associations of serum leptin levels during the course of infectious diseases may differ in men and women in regard to disease susceptibility, progression, and outcome.

Leptin levels were described to be higher in COVID-19 pneumonia patients compared to healthy controls. This difference was observed in patients with a BMI of ≥25 kg/m^2^, but not in those with a lower BMI [71]. Several other studies also reported higher serum leptin levels in COVID-19 patients compared to healthy controls [73,108,118,119]. Yet, in these studies, BMI either did not differ between the two cohorts or was not evaluated [73,108,118,119].

In 62 COVID-19 patients whose symptom onset was less than 3 days before hospital admission, circulating leptin levels were elevated in comparison to 62 healthy controls. Although this adds to the reports pointing to elevated leptin levels being indicative of a SARS-CoV-2 infection; the implications of this study are limited due to the higher BMI of the COVID-19 patients, which may well explain higher serum leptin levels irrespective of the SARS-CoV-2 infection [74]. Hence, it remains to be clarified if circulating leptin levels are associated with BMI in SARS-CoV-2-infected patients. Furthermore, adding to the uncertainty in the field, another study could not detect major differences in plasma leptin levels between healthy controls and patients with mild, severe, and critical COVID-19 [76]. Likewise, and in contrast to the studies described above [71,73,108,118,119], systemic leptin levels were comparable in patients with mild, moderate, and severe COVID-19 disease [68]. Corroborating the latter results, in a cohort of 254 COVID-19-infected patients, serum leptin levels were similar in patients with severe and non-severe disease on the day of admission and on Days 7 and 28 after hospitalization. Notably, serum leptin amounts declined in both groups during hospital stay [63], which may indicate that serum leptin levels normalize during the course of the disease.

However, patients with severe COVID-19 were also reported to have approximately 3-fold higher serum leptin levels compared to critical COVID-19 patients [76]. While this may suggest that serum leptin concentrations decline in advanced COVID-19 disease, this finding was not related to survival in critical COVID-19 patients [76]. Adding further complexity to the interpretation of studies determining leptin levels in COVID-19 patients, logistic regression analysis of data from 128 COVID-19 patients demonstrated that leptin levels were protective, and higher serum leptin concentrations were associated with lower in-hospital mortality [120].

On the other hand, serum leptin levels were not related to ARDS or outcome [108] and did not correlate with the lung ultrasound score [74]. In a group of 145 hospitalized patients, leptin levels were not related to intensive care requirement or patient outcome [69]. Likewise, in a cohort of 195 hospitalized COVID-19 patients plasma leptin levels remained rather constant during hospital stay and were not related to inflammation or clinical outcomes [67]. A prospective study with 195 hospitalized COVID-19 patients did not observe different plasma leptin levels between patients not admitted and those admitted to the intensive care unit, and between survivors and non-survivors [67]. Similarly, monitoring COVID-19 patients over time in a cohort in Brazil, leptin levels were unrelated to disease severity and mortality [72]. Additionally, plasma leptin amounts were not associated with intensive care unit admission and/or length of stay [73]. Yet, in a cohort of 31 COVID-19 patients from China, leptin levels predicted disease severity and were associated with disease progression. Within this patient group, high leptin levels were related to the low lymphocyte number and M1 polarization of macrophages [119].

Some studies also compared leptin levels between comparably ill patients with and without COVID-19. Here, leptin levels were similar in patients with COVID-19 and non-COVID-19 suffering from severe pneumonia, and were neither related to the inflammatory cytokines measured or the disease outcome [65]. In spite of this, other reports still allow for different interpretations, as serum leptin levels were almost 4-fold higher in SARS-CoV-2-related respiratory failure compared to critically ill ventilated patients not infected by this virus [121].

One study also addressed the significant number of COVID-19 patients experiencing gastrointestinal and/or intestinal symptoms [122]. However, serum leptin levels were not changed in patients with intestinal lesions or hepatitis compared to patients without gastrointestinal complications [108].

In summary, the identification of general concepts in relation to leptin levels in COVID-19 disease severity and outcome remains difficult, mainly due to the often limited statistical power of small and heterogeneous cohorts in relation to age, gender, and BMI. Other variables of currently available data include different SARS-CoV-2-strain infections, vaccination status, treatment regimens, and possibly additional underlying medical conditions. Nevertheless, several studies support a trend towards elevated leptin levels in COVID-19 patients. It remains to be clarified if these observations are related to disease severity. Additionally, leptin levels appear to not suitable to serve as a prognostic marker for admission to the intensive care unit or survival (Table 1).

Yet, more valuable information regarding the potential of leptin to serve as a diagnostic and prognostic marker might be obtained from the adiponectin/leptin ratio, which is an excellent marker for metabolic diseases, and negatively correlates with measures of low-grade inflammation [123]. In fact, a high adiponectin-to-leptin ratio was related to a state of good health with lower BMI, less diabetes and hypertension, a lower risk for admission to the intensive care unit, and death in cohorts of SARS-CoV-2-infected patients. In these studies, and indicating marker potential, the adiponectin/leptin ratio enabled discrimination of COVID-19 pneumonia and healthy controls [68,71] (Table 1).

### 2.4. Resistin

Resistin is an adipocyte-derived protein in rodents, and it is elevated in obese rodent animals. In contrast, in humans, resistin is primarily released by monocytes/macrophages. Inflammatory cytokines or LPS can induce resistin expression in monocytes/macrophages, and as a result, resistin was designated as a pro-inflammatory factor associated with low-grade inflammation in obesity [124,125]. While T and B cells express resistin only at low levels [126], neutrophils store significant amounts of resistin in granules, which is released by stimulation with the chemotactic peptide N-formylmethionyl-leucyl-phenylalanine, as well as several bacterial compounds [126,127]. Activities of resistin to upregulate inflammatory reactions include its ability to enhance the release of pro-inflammatory cytokines in THP-1 monocytes and peripheral blood mononuclear cells. On the other hand, more recent research indicated that resistin functions as an anti-microbial protein, an anti-inflammatory molecule in particular infections, and contributes to the resolution of inflammation [126].

Circulating resistin levels were found to be increased in patients with viral infections. In patients infected with Puumala hantavirus, plasma resistin concentrations were induced in the acute phase and declined in the recovery period. These patients often suffer from acute kidney injury, and plasma resistin levels were related to urinary albumin levels [128]. In HBV-infected patients, the serum resistin amounts were higher compared to healthy controls. Further pointing to resistin levels reflecting liver function, serum resistin concentrations were induced in patients with liver cirrhosis or liver failure [129]. Less increase in serum resistin levels was found in HCV compared to HBV-infected patients [130]. However, an association of serum resistin amounts with histological parameters of liver disease severity could not be identified in patients suffering from chronic HCV infections [131]. Hence, the suitability of systemic resistin levels as a non-invasive marker for liver diseases needs further study.

Nevertheless, a therapeutic potential of resistin may exist, as it was found to regulate the expression of IFN λ-3. This observation appears highly relevant for the success of IFN therapy for HCV-infected patients prior the establishment of direct-acting antiviral therapy. In support of this, high pre-treatment resistin levels were associated with a better response to IFN therapy [126,132,133].

Plasma resistin levels in COVID-19 patients are elevated compared to healthy controls [74,75]. One of these studies compared resistin levels between COVID-19 and sepsis patients, and showed increased levels in the latter group. The Acute Physiology and Chronic Health Evaluation (APACHE) II score and the Sequential Organ-Failure Assessment (SOFA) score were also higher in sepsis, and increased serum resistin concentrations seem to resemble a marker of disease severity independent of a SARS-CoV-2 infection [75]. Indeed, an association of resistin levels with the SOFA score in patients with sepsis was also shown by others [134].

Higher resistin levels were detected in 159 severely ill and 71 critically ill COVID-19 patients, compared to 30 patients with mild symptoms [76]. Plasma resistin amounts were markedly induced in COVID-19 patients admitted to the intensive care unit in comparison to less ill patients and asymptomatic controls. This study also provided evidence that resistin released from neutrophils was the main source of elevated plasma resistin levels. Strikingly, and showing marker potential for disease progression and outcome, high plasma resistin levels of non-critically ill COVID-19 patients were characteristic for those patients that progressed to severe illness and were less likely to survive the COVID-19 disease [76]. In line with this finding, data of 3325 SARS-CoV-2-infected patients revealed high numbers of immature granulocyte and neutrophils, but not monocytes, at early disease stages to be associated with mortality [135].

In a group of 146 patients, which was stratified for pneumonia severity in mild, moderate, and severe illness, serum resistin levels were almost 2-fold higher in the latter group. An increase of IL-6, TNF, CXCL8, and CCL2 was also noticed in the severely ill patients. With the exception of CCL2, all of these cytokines and chemokines declined 4–6 weeks after diagnosis. Showing marker potential, serum resistin levels were predictors of the prognosis of pneumonia severity and the need for invasive mechanical ventilation [77]. However, these resistin-based predictions were unrelated to obesity and metabolic syndrome [77]. Ebihara and colleagues provided additional evidence for the association of high circulating resistin levels with COVID-19 disease severity and survival [75]. Day 1 resistin levels were related to the disease severity score, which was defined as the maximum acuity score during the study (A1, dead; A2, intubated, survived; A3, hospitalized with oxygen; A4, hospitalized without oxygen; A5, discharged). In these studies, serum resistin levels measured on Day 1, 4, and 8 were higher in non-survivors than survivors. In addition, high plasma resistin concentrations in the early course of the COVID-19 disease were related to late recovery [75].

Corroborating that resistin levels could be a useful marker for COVID-19 disease outcome, it was identified in a cohort of 254 COVID-19-infected patients that serum resistin levels were higher in patients with severe disease compared to those patients with non-severe disease on the day of admission. This relationship between resistin levels and disease outcome was not apparent on Day 7 and Day 28 after hospitalization. Notably, serum resistin levels declined during hospital stay in patients with severe and moderate disease and did not significantly change in those with mild symptoms [63].

However, the exact time point to determine resistin levels for the prognosis of the COVID-19 disease progression has yet to be determined, as another study determined that only the plasma resistin levels of COVID-19 patients measured on Days 4–6 after hospital admission, but not earlier, were related to survival. The circulating resistin of normal weight, overweight, and obese patients was similar [67], which supports the data sets described above [77] and makes it unlikely that resistin levels reflect the link between obesity and COVID-19 disease severity. The expression pattern of resistin in humans, being predominantly released by monocytes/macrophages activated by inflammatory cytokines, might be a critical contributor to this observation. Nevertheless, circulating resistin levels appear upregulated in SARS-CoV-2 infection, and within a period during the early stages of COVID-19, which is yet to be better defined, high resistin levels are related to disease severity and may serve as a prognostic marker for admission to the intensive care unit and survival (Table 1).

### 2.5. Galectin-3

Galectin-3 is found in multiple intra- and extracellular locations, including the cytoplasm and nucleus, as well as upon secretion, in the extracellular milieu. Outside cells, galectin-3 binds beta-galactoside residues of a large variety of proteins [136]. Intra- and extracellular galectin-3 have multiple roles in cell proliferation, apoptosis, angiogenesis, and cell differentiation [137,138]. Most relevant for this review, galectin-3 also influences inflammatory processes and viral infection [137,138]. Moreover, circulating galectin-3 levels were found to be higher in the obese, and were associated with metabolic diseases [136,139], implying a potential link between galectin-3 levels in virus-related infections and inflammatory events in obesity.

Plenty of evidence points to galectin-3 enhancing inflammation and fibrosis in tissues such as in the liver and lungs [137,140]. Galectin-3 is a chemotactic protein for monocytes and macrophages, which themselves express high levels of galectin-3. This lectin can activate and stimulate oxidative burst in neutrophils, which is supposed to contribute to the cytokine storm in COVID-19, resulting in more severe disease [137,141]. Chronic inflammation triggers the development of tissue fibrosis, and galectin-3 was shown to activate tissue myofibroblasts enhancing their synthesis of extracellular matrix proteins [137], which, when extensive, furthers fibrosis. Along these lines, galectin-3 contributes to pulmonary fibrosis, which is a relatively common and very serious complication of COVID-19 disease [142].

Most relevant for the COVID-19 disease progression, galectin-3 is also known to often worsen viral infectious diseases. Prominent examples for galectin-3-mediated and infection-related activities include its ability to facilitate herpes simplex virus cell entry and to induce the death of human immunodeficiency virus (HIV)-infected macrophages. Galectin-3 was also shown to upregulate antiviral genes to inhibit influenza A virus replication [138]. Moreover, galectin-3 expression is induced in immune cells infected by HIV or human T lymphotropic virus type I [138] and upregulated in the serum of HCV-infected patients [138,143]. While the consequences of infection-related elevation of galectin-3 levels remain to be fully understood, galectin-3 upregulation enhanced inflammatory cytokine levels and myofibroblast activation, both with roles in lung fibrosis [144]. With pulmonary fibrosis in ARDS representing a life-threating outcome of SARS-CoV-2 infection and serum galectin-3 levels predicting survival in ARDS [144], galectin-3 levels were also measured in COVID-19 patients.

Strikingly, serum galectin-3 levels were higher in COVID-19 patients compared to controls, and could discriminate non-survivors and survivors [79,82,85,90,145]. For instance, 29 severe COVID-19 cases displayed higher serum galectin-3 levels than 55 non-severe cases [85]. In line with this, stratification of 280 COVID-19 patients for disease severity (mild, moderate, severe, and critical) showed a consistent rise of circulating galectin-3 levels from a mild to a critical state [83]. Single-cell analysis found elevated levels of galectin-3 in monocytes, macrophages, and dendritic cells of patients with severe COVID-19 as compared to mild cases [81]. Serum galectin-3 concentrations were an independent predictor of severe outcome, which was defined as the need for invasive mechanical ventilation [82] and was also a predictor of severe pneumonia [78,84,86]. Furthermore, galectin-3 levels at admission were risk factors for the need of intensive care and advanced ventilatory support [87]. Likewise, higher galectin-3 levels at hospital admission were related to intensive care unit stay or death at 30 days [80,88]. Associations of higher plasma galectin-3 levels at baseline in COVID-19 patients with worse outcomes were also reported by others [89]. Although galectin-3 levels did not predict unfavourable outcomes in this cohort (*n* = 358), it was an independent predictor of 30-day mortality [89].

Interestingly, in another study, elevated galectin-3 levels also predicted admission to intensive care, but this was not restricted to COVID-19, but patients with respiratory viral infection, bacterial pneumonia or non-infectious inflammatory diseases [80] were also included, indicating that serum galectin-3 levels have a much broader potential as a prognostic marker beyond COVID-19.

High galectin-3 levels in patients with severe COVID-19 led Gaughan and coworkers to propose the pharmacological galectin-3 inhibitor GB0139 as a potential therapeutic approach in COVID-19-related pneumonitis. The randomized controlled clinical trial showed that GB0139 inhalation achieved clinically relevant GB0139 levels in plasma and was well-tolerated by patients with COVID-19 pneumonitis [146]. The outcomes on the impact of GB0139 on COVID-19 severity in this trial are yet to be described.

Importantly, one study contradicted the trend of upregulated galectin-3 levels correlating with the COVID-19 disease outcome described above. This prospective case–control study included 100 patients, and lower serum galectin-3 levels were measured in severe/critical COVID-19 patients compared to the moderate disease group and healthy controls. Nevertheless, galectin-3 levels predicted admission to the intensive care unit with 75% sensitivity and 50% specificity [147].

Taken together, and with only one exception, circulating galectin-3 levels appear elevated in COVID-19 infection, and high galectin-3 levels are related to disease severity. The findings could make galectin-3 a valuable marker to prognose admission to the intensive care unit and survival in SARS-CoV-2-infected patients (Table 1).

### 2.6. Visfatin

Visfatin, which is also named the pre-B cell colony enhancing factor or nicotinamide phosphoribosyltransferase, produces nicotinamide mononucleotides important for nicotinamide adenine dinucleotide (NAD) biosynthesis [148,149]. Given the large and diverse number of enzymes that require NAD for their activity, regulating an enormous variety of cellular functions, intracellular visfatin can modulate numerous pathways. In addition, and most relevant in the context of this review, visfatin released from cells was shown to enhance inflammation and fibrosis [150,151]. Moreover, visfatin expression in adipose tissues was found to be induced in the obese, and accordingly, circulating visfatin levels were also higher. Likewise, visfatin levels in plasma were also increased in patients with metabolic diseases, such as type 2 diabetes [152,153].

Visfatin levels were found upregulated in the lung tissues of experimental ARDS mice. Implicating therapeutic potential through the inhibition of visfatin’s inflammatory mode of action, the infusion of visfatin neutralizing antibodies improved lung injury and attenuated inflammation [154]. Such a therapeutic approach was also effective in rat and porcine ARDS models [155].

In COVID-19 patients, visfatin levels were high in critically ill patients in comparison to patients with severe and mild disease. Patients admitted to the intensive care unit not infected with SARS-CoV-2 had lower visfatin serum levels compared to similarly ill patients infected by this virus [76]. Based on the findings derived from experimental ARDS models in mice, rats, and porcine, blockage of visfatin activity via injection of inhibitory antibodies may have potential to treat severely ill COVID-19 patients.

### 2.7. Apelin

Apelin is a 77 amino acid pre-pro-peptide, and upon multiple cleavages, several different active apelin peptide isoforms in plasma exist, including apelin-36 and apelin-17, all of which signal through the apelin receptor APJ to confer cardioprotective properties [156,157].

While angiotensin-converting enzyme 2 (ACE2) has been recognized as a cell surface receptor through which SARS-CoV-2 can enter host cells, this enzyme also contributes to the regulation of the renin–angiotensin–aldosterone (RAS) system that controls blood pressure and fluid and electrolyte homeostasis. Its carboxypeptidase activity removes one C-terminal amino acid from angiotensin-II, thereby lowering the levels of the peptide that critically determines RAS activity to generate angiotensin [158]. In addition, ACE also cleaves apelin peptides, thereby lowering their beneficial cardiovascular activities [159]. In addition, in vitro data demonstrated that apelin-13 protects bronchial epithelial cells from SARS-CoV-2 infection, and also lowers the inflammatory responses to viral infection [160]. These protective properties appear reduced in experimental ARDS, where lower circulating and lung-expressed apelin levels were observed [161].

Apelin improves glucose and fatty acid metabolism, and protects from obesity, hypertension and cardiovascular diseases. In several study cohorts, serum apelin levels were elevated in obesity, hypertension, and type 1 and type 2 diabetes [156,162]. In fact, high apelin levels in the obese may be considered protective, as this protein can enhance the browning of white adipocytes [157]. As brown adipose tissue can burn calories [163], these findings suggested that apelin upregulation may contribute to body weight loss. Yet, another study did not confirm these implications and rather documented reduced apelin levels in obesity and metabolic diseases [164,165].

Serum apelin levels were lower in COVID-19 patients compared to healthy controls [79]. COVID-19 infections lowered serum apelin concentrations in type 2 diabetes and hypertension. On the other hand, higher serum apelin levels correlated with an improved arterial oxygen saturation and less severe lung involvement [166], but not survival [79]. Although these studies may point at a diagnostic value to correlate serum apelin levels with infection and disease, the relevant cohorts consisted of 69 and 78 COVID-19 patients [79,166], respectively, and future research with larger cohorts will be needed to substantiate these findings.

## 3. Conclusions

Obesity increases the susceptibility and risk for SARS-CoV-2 infections, a worse COVID-19 disease trajectory and COVID-19-related death [13]. Circulating levels of chemerin, leptin, and galectin-3 are commonly elevated, while serum adiponectin levels are reduced in the obese and all of those changes are related to metabolic diseases (Figure 2) [5,28,43]. While a constantly increasing number of studies have determined adipokine levels in COVID-19 patients, a major limitation of currently available information is the lack of data comparing COVID-19 patients with similarly ill patients not infected by this virus. Therefore, changes related to COVID-19 severity and outcome cannot be easily discriminated.

Based on the publicly available patient data, it appears that there is little evidence to support diagnostic or prognostic potential for changes in adiponectin and chemerin levels in COVID-19 disease. Systemic leptin levels were found to be increased in SARS-CoV-2-infected patients, but associations with COVID-19 disease severity were not consistently observed [65,74]. However, leptin was shown to induce expression of resistin and galectin-3, and thereby may contribute to higher levels of resistin and galectin-3 in the serum of COVID-19 patients (Figure 2) [167,168].

Furthermore, the adiponectin/leptin ratio of COVID-19 patients is low, and may serve as a tool to predict admission to intensive care and death [68,71]. The association of the adiponectin/leptin ratio with adipose tissue dysfunction and metabolic diseases were described [123], and current findings in COVID-19 disease encourage prospective studies to further evaluate this ratio as a clinical relevant indicator. Resistin and galectin-3 may emerge as biomarkers of COVID-19 disease activity and prognosis [77,83] (Figure 2). Adipokines, such as apelin and visfatin, are less well-studied, and their potential as disease markers for COVID-19 is still to be clarified. Based on the current knowledge on adipokine functions, it appears unlikely that changes in the systemic levels of chemerin, leptin, and adiponectin, which are mainly released by the adipocyte, have a significant direct role in COVID-19 infection and disease course. These proteins are, however, related to metabolic dysfunction, such as insulin resistance and hypertension, known to increase the risk of severe COVID-19 [3,4,24]. On the other hand, resistin and galectin-3 are expressed by adipocytes, but are much more abundant in immune cells [124,137,139], indicating that altered immune cell functions in the lung, other organs, and probably also adipose tissues are major contributors for a worse disease outcome during the course of COVID-19. The cytokine storm in severe COVID-19 masks obesity-related inflammatory processes [77], suggesting that the adipokine profile in early disease and/or during the disease course may be different in the obese.

## Figures and Tables

**Figure 1 biomedicines-11-01302-f001:**
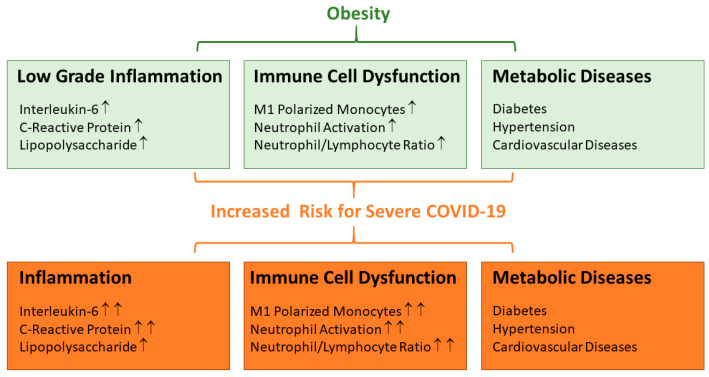
Low-grade inflammation, immune cell dysfunction, and metabolic diseases are associated with obesity. All of these features are also risk factors for a more severe outcome of COVID-19. In COVID-19 patients, inflammation with higher circulating levels of interleukin-6, C-reactive protein, and lipopolysaccharide is associated with a more severe course of the disease. Elevated M1 macrophages, activated neutrophils, and increased neutrophil/lymphocyte ratio contribute to disease severity. Metabolic disorders such as diabetes, hypertension, and cardiovascular disease are risk factors for a severe COVID-19 disease outcome (see text for details). ↑ Increased in the circulation, ↑↑ Strongly increased in the circulation.

**Figure 2 biomedicines-11-01302-f002:**
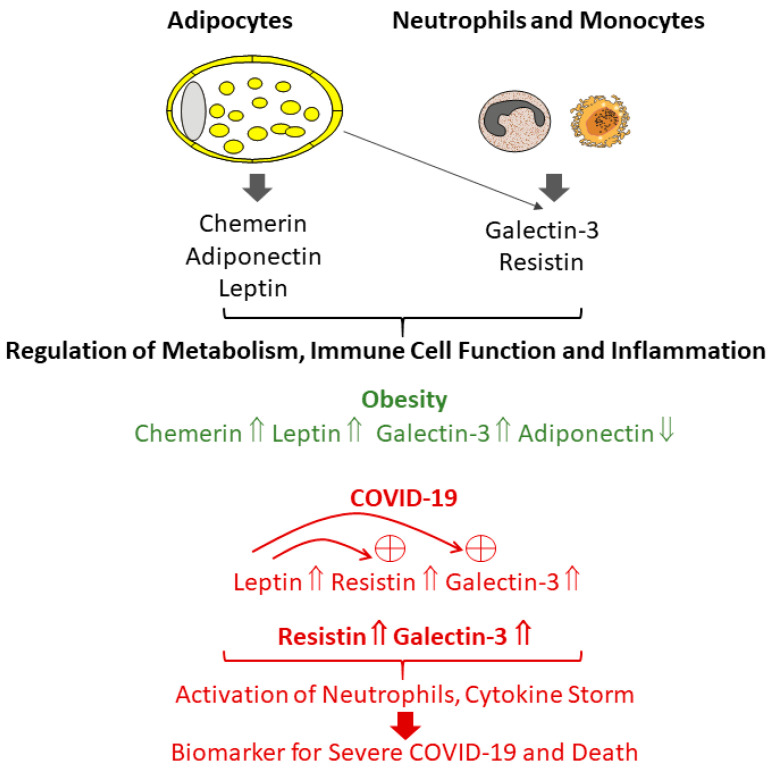
Chemerin, adiponectin and leptin are mostly released by adipocytes, while galectin-3 and resistin are secreted from monocytes and neutrophils. All of these adipokines regulate metabolism, immune cell function, and inflammation. Chemerin, leptin, and galectin-3, but not resistin levels, are increased in the serum of the obese. This partially matches observations in COVID-19 patients, which display elevated leptin, resistin, and galectin-3 serum levels. As indicated, crosstalk between adipokines exists, and leptin may further induce resistin and galectin-3 expression and secretion, suggesting that adipocyte-derived cytokines may have a role in the response of macrophages and neutrophils to SARS-CoV-2 infection and COVID-19 severity. In particular, high resistin and galectin-3 levels are related to neutrophil activation and the cytokine storm, and are biomarkers for SARS-CoV-2-related severe COVID-19 disease and fatal outcome, death.

**Table 1 biomedicines-11-01302-t001:** The different adipokines analysed in serum/plasma of COVID-19 patients are listed. Their levels compared to healthy controls, and correlations or trends of adipokine levels related to (i) disease severity, (ii) prognosis for severe disease or (iii) death are indicated. The table refers to major trends observed in multiple studies. Inconsistent data or lack thereof are indicated (?).

Adipokine	Elevated in COVID-19	Related to Disease Severity	Prognostic for Admission to Intensive Care	Prognostic for Survival	References
Chemerin	?	No	No	No	[62,63]
Adiponectin	No	No	No	No	[65,66,67,68,69,70,71]
Leptin	Yes	No	No	No	[63,67,68,69,71,72,73,74]
Adiponectin/Leptin	Yes	?	Yes	Yes	[68,71]
Resistin	Yes	Yes	Yes	Yes	[63,67,74,75,76,77]
Galectin-3	Yes	Yes	Yes	Yes	[75,78,79,80,81,82,83,84,85,86,87,88,89,90]

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
