# Peer review of "Adipokines as Diagnostic and Prognostic Markers for the Severity of COVID-19"

_biomedicines, 2023, doi:10.3390/biomedicines11051302_

Round 1

Reviewer 1 Report

The authors aimed to correlate the circulating levels of adipokines with progression and disease outcome of COVID-19. This paper increases our knowledge and broad overview about the correlation between adipokines and the severity of COVID-19. Overall, the manuscript is well-written and well-organized.

Minor concerns:

1. Table 1: The different adipokines analyzed in COVID-19 patients are listed. Please cite relevant references.

2. Recent published articles on this topic (Int J Obes. 2023 Feb;47(2):126-137, Int J Mol Sci. 2023 Jan 6;24(2):1131). The authors should cite the references and discuss in the section of discussion.

Author Response

We are very grateful to the reviewer for the kind comments on our review article

Minor concerns:

  1. Table 1: The different adipokines analyzed in COVID-19 patients are listed. Please cite relevant references.

References are now included in the last column of the table.

  1. Recent published articles on this topic (Int J Obes. 2023 Feb;47(2):126-137, Int J Mol Sci. 2023 Jan 6;24(2):1131). The authors should cite the references and discuss in the section of discussion.

These two articles were cited (please see references 74 and 76) and discussed.

74        Perrotta F.; Scialo F.; Mallardo M.; Signoriello G.; D'Agnano V.; Bianco A.; Daniele A.; Nigro E. Adiponectin, Leptin, and Resistin Are Dysregulated in Patients Infected by SARS-CoV-2. Int J Mol Sci 2023, 24. https://doi.org/10.3390/ijms24021131.

76        Flikweert A. W.; Kobold A. C. M.; van der Sar-van der Brugge S.; Heeringa P.; Rodenhuis-Zybert I. A.; Bijzet J.; Tami A.; van der Gun B. T. F.; Wold K. I.; Huckriede A.; Franke H.; Emmen J. M. A.; Emous M.; Grootenboers M.; van Meurs M.; van der Voort P. H. J.; Moser J. Circulating adipokine levels and COVID-19 severity in hospitalized patients. Int J Obes (Lond) 2023, 47, 126-137. https://doi.org/10.1038/s41366-022-01246-5.

Reviewer 2 Report

The manuscript is interesting and shed new light on a poorly investigated research field, highlighting a potential and interesting role for adipocyte-derived signals in influencing overall health status and especially COVID-19 disease evolution.

The review clearly explain the major physiological role of the investigated molecules and highlight the current evidence for their investigation in COVID-19 field, highlighting, at the same time, the limitations of the current knowledge about some of them.

The work sounds interesting and only minor revisions are required, especially to fix some typesetting errors. Moreover, please carefully revise the sentence at page 8, lines 377-379 in the leptin section: in my opinion there is a mistake in the sentence construction, as in the present form it make no sense.

Author Response

We are very grateful to the reviewer for the kind comments on our review article.

We corrected the manuscript and hopefully found all of the typesetting errors. The sentence at page 8 was corrected “Strikingly, and identifying leptin as a key player controlling the immune system, treatment with recombinant leptin normalized the immune response”